# Aligning Model Properties via Conformal Risk Control

**William Overman**[1]    **Jacqueline Jil Vallon**[2]    **Mohsen Bayati**[1]

[1] Stanford Graduate School of Business    [2] Management Science and Engineering
{wpo,bayati}@stanford.edu,    jjvallon@alumni.stanford.edu

## Abstract

AI model alignment is crucial due to inadvertent biases in training data and the underspecified machine learning pipeline, where models with excellent test metrics may not meet end-user requirements. While post-training alignment via human feedback shows promise, these methods are often limited to generative AI settings where humans can interpret and provide feedback on model outputs. In traditional non-generative settings with numerical or categorical outputs, detecting misalignment through single-sample outputs remains challenging, and enforcing alignment during training requires repeating costly training processes. In this paper we consider an alternative strategy. We propose interpreting model alignment through property testing, defining an aligned model $f$ as one belonging to a subset $\mathcal{P}$ of functions that exhibit specific desired behaviors. We focus on post-processing a pre-trained model $f$ to better align with $\mathcal{P}$ using conformal risk control. Specifically, we develop a general procedure for converting queries for testing a given property $\mathcal{P}$ to a collection of loss functions suitable for use in a conformal risk control algorithm. We prove a probabilistic guarantee that the resulting conformal interval around $f$ contains a function approximately satisfying $\mathcal{P}$. We exhibit applications of our methodology on a collection of supervised learning datasets for (shape-constrained) properties such as monotonicity and concavity. The general procedure is flexible and can be applied to a wide range of desired properties. Finally, we prove that pre-trained models will always require alignment techniques even as model sizes or training data increase, as long as the training data contains even small biases.

## 1 Introduction

The emergence of large foundation models has increased the attention to the problem of alignment. Aligned models are artificial intelligences designed to pursue goals that align with human values, principles, and intentions (Leike et al., 2018; Ouyang et al., 2022; Hendrycks et al., 2023; Ngo et al., 2024). Although the alignment problem is predominantly examined in the context of potential artificial general intelligence (AGI), large language models (LLMs), and reinforcement learning (RL) agents, it also has roots in the modern machine learning pipeline (D'Amour et al., 2022). Motivated by this, we introduce a broader notion of alignment in this paper, extending beyond the aforementioned generative models to include even tabular regression models.

As an example, consider a regression task where property $\mathcal{P}$ represents models that are monotonically decreasing in a given feature (covariate). For example, predicting cancer patient survival should be monotonically decreasing in cancer stage (Vallon et al., 2022, 2024). Constraining a prediction model during training to maintain monotonicity in this feature can be viewed as a form of alignment. For a pre-trained model $f$ that was trained without such constraints, ensuring monotonically decreasing predictions in this feature can be more complex. This complexity arises particularly in non-generative

settings where the user cannot update $f$ or obtain any outputs other than point predictions $f(X)$ for a given input $X$.

In this work, we propose an approach to aligning a pre-trained model $f$ that is motivated by property testing (Ron, 2008; Goldreich, 2017) and conformal risk control (Angelopoulos et al., 2024). Property testing aims to design efficient algorithms for determining membership to the set $\mathcal{P}$ of functions with a given property, that require fewer resources than learning algorithms for $\mathcal{P}$ (Ron, 2008). This is particularly relevant for modern deep learning, where a user may need to determine if a pre-trained model $f$ belongs to $\mathcal{P}$ without the resources to train a model of comparable size.

Property testing algorithms use local queries to determine, with high probability, whether a function has a given global property or is far from having it. We map such queries for a property $\mathcal{P}$ to a set of loss functions, which we then use in a conformal risk control procedure (Angelopoulos et al., 2024) to establish a notion of alignment for $\mathcal{P}$. We prove that this procedure yields a conformal interval around $f$ containing a function close to $\mathcal{P}$.

We demonstrate our methodology on real-world datasets for the properties of monotonicity and concavity. Motivated by the potential for systematic under- or over-estimation bias in $f$, we provide a straightforward extension of Angelopoulos et al. (2024) to obtain asymmetric conformal intervals with multi-dimensional parameters. While we examine both monotonicity and concavity constraints, the majority of our focus is on monotonicity, as these constraints have been shown to promote crucial aspects of alignment to human values, such as fairness and adherence to social norms (Wang and Gupta, 2020).

While our methodology provides a way to align pre-trained models, one may question whether such techniques will remain necessary as AI capabilities advance. Given the outstanding capabilities of modern AI models with substantially large numbers of parameters and training data, one may argue that the alignment problem may naturally disappear as such advances continue (Kaplan et al., 2020). However, another contribution of this paper is to refute this argument in a stylized setting, building on recent advances in the theory of linearized neural networks (Mei and Montanari; Misiakiewicz and Montanari, 2023). Specifically, we show that increasing the size of the training data or the number of parameters in a random feature model (a theoretically tractable neural network proxy where hidden layer weights are randomly initialized and fixed (Rahimi and Recht, 2007)) cannot help it satisfy a property $\mathcal{P}$, if the pre-training data has biased labels. Our simulations show that the result holds even if only a small fraction of the training labels are impacted by the bias.

Summarizing our main contributions, we: (1) introduce an alignment perspective based on property testing, (2) use conformal risk control to post-process predictions of pre-trained models for better alignment, and (3) demonstrate that increasing training data and parameters in a random feature model does not eliminate the need for alignment. We discuss related work in Section 6, particularly our connections to Yadkori et al. (2024), who use conformal risk control to address large language model hallucinations (Ji et al., 2023).

## 2 Preliminaries

In this section we provide key definitions drawn from property testing as well as a condensed overview to conformal prediction and conformal risk control. We provide a short introduction to propery testing in the extended version of the paper Overman et al. (2024) and an extensive introduction to the field can be found in Goldreich (2017).

### 2.1 Properties and Property Testing for Set-Valued Functions

Our perspective on alignment in this work is motivated by the field of property testing (Goldreich, 2017; Ron, 2008). Property testing studies algorithms that, by making a small number of local queries to a large object (such as a function or a graph), can determine whether the object has a certain property or is significantly far from having it.

Classic examples include linearity testing of Boolean functions (Blum et al., 1993), testing whether a function is a low-degree polynomial (Kaufman and Ron, 2006; Bhattacharyya et al., 2009), and testing $k$-juntas (Blais, 2009). These algorithms generally operate by randomly sampling and querying the object, leveraging local information to infer global properties.

In this work, we focus on *set-valued functions*, which are functions that map elements of a domain $\mathcal{X}$ to subsets of a codomain $\mathcal{Y}$, i.e., $F : \mathcal{X} \to 2^{\mathcal{Y}}$. While the standard definitions of property testing are technically sufficient for our purposes—since we can consider set-valued functions as functions with range $2^{\mathcal{Y}}$—we introduce specialized definitions to maintain clarity and to facilitate the transition between discussing $\mathcal{Y}$ and $2^{\mathcal{Y}}$.

**Definition 1** (Satisfying and Accommodating a Property). *Let* **property** $\mathcal{P}$ *denote a specific subset of all functions that map $\mathcal{X}$ to $\mathcal{Y}$. A function $f : \mathcal{X} \to \mathcal{Y}$ **satisfies** the property $\mathcal{P}$ if $f \in \mathcal{P}$.*

*A set-valued function $F : \mathcal{X} \to 2^{\mathcal{Y}}$ **accommodates** a property $\mathcal{P}$ if there exists a function $g \in \mathcal{P}$ such that $g(x) \in F(x)$ for all $x \in \mathcal{X}$.*

Intuitively, $F$ accommodates $\mathcal{P}$ if it contains at least one function $g$ satisfying $\mathcal{P}$ within its possible outputs.

We extend the notion of $\varepsilon$-farness from a property (as defined in the extended version of the paper Overman et al. (2024)) to set-valued functions. For set-valued functions, we measure the distance based on how often the outputs of any function $g \in \mathcal{P}$ fall within the sets provided by $F$.

**Definition 2** ($\varepsilon$-Faraway). *For a set-valued function $F : \mathcal{X} \to 2^{\mathcal{Y}}$, a distribution $\mathcal{D}$ over $\mathcal{X}$, $\varepsilon > 0$, and a property $\mathcal{P}$, we say $F$ is $\varepsilon$-**Faraway** from $\mathcal{P}$ with respect to $\mathcal{D}$ if $\delta_{\mathcal{P},\mathcal{D}}(F) > \varepsilon$, where*

$$\delta_{\mathcal{P},\mathcal{D}}(F) \overset{def}{=} \inf_{g \in \mathcal{P}} \delta_{\mathcal{D}}(F, g) \quad and \quad \delta_{\mathcal{D}}(F, g) \overset{def}{=} \Pr_{X \sim \mathcal{D}}[g(X) \notin F(X)].$$

**Note.** Throughout this work, we assume that $\mathcal{D}$ is the empirical distribution of a fixed and finite calibration dataset, and thus has finite support. While this assumption is not strictly necessary, most property testing results are over finite domains. Property testing over functions with Euclidean domains is a in general a difficult problem, though there have been notable recent successes (Fleming and Yoshida, 2020; Arora et al., 2023)..

With these definitions in place, we can define *testers* for set-valued functions. We focus on *one-sided error testers*, which are algorithms that take in a set-valued function $F$, a distribution $\mathcal{D}$, and a distance parameter $\varepsilon$ and output either Accept or Reject. These algorithms never reject a function that accommodates the property. The standard definition of one-sided error testers (provided in the extended version of the paper Overman et al. (2024)) extends naturally to set-valued functions by replacing the notion of satisfying a property with accommodating it.

**Definition 3** (One-Sided Error Tester for Set-Valued Functions). *A one-sided error tester for a property $\mathcal{P}$ in the context of set-valued functions is a probabilistic oracle machine $\mathcal{M}$ that, given a distance parameter $\varepsilon > 0$, oracle access to a set-valued function $F : \mathcal{X} \to 2^{\mathcal{Y}}$, and oracle access to samples from a fixed but unknown distribution $\mathcal{D}$ over $\mathcal{X}$, satisfies:*

  1. *If $F$ accommodates $\mathcal{P}$, then $\Pr[\mathcal{M}^{F,\mathcal{D}}(\varepsilon) = Accept] = 1$.*

  2. *If $F$ is $\varepsilon$-Faraway from $\mathcal{P}$ with respect to $\mathcal{D}$, then $\Pr[\mathcal{M}^{F,\mathcal{D}}(\varepsilon) = Accept] \leq \frac{1}{3}$.*

*Here, $\mathcal{M}^{F,\mathcal{D}}(\varepsilon)$ denotes the execution of the tester $\mathcal{M}$ when given oracle access to the function $F$, the distribution $\mathcal{D}$, and the parameter $\varepsilon$.*

Note that $\mathcal{M}$ itself is an abstract algorithm; $\mathcal{M}^{F,\mathcal{D}}$ is the instantiation of this algorithm with specific oracle access to $F$ and $\mathcal{D}$.

In many property testing algorithms, the parameter $\varepsilon$ is used only to determine the number of iterations or samples required, not the core logic of the tester. This leads to the concept of *proximity-oblivious testers* (POTs), where the basic testing procedure is independent of $\varepsilon$. The general definition of POTs (given in the extended version of the paper Overman et al. (2024)) also extends naturally to set-valued functions.

**Definition 4** (Proximity-Oblivious Tester for Set-Valued Functions). *A proximity-oblivious tester for a property $\mathcal{P}$ in the context of set-valued functions is a probabilistic oracle machine $\mathcal{T}$ that satisfies:*

  1. *If $F$ accommodates $\mathcal{P}$, then $\Pr[\mathcal{T}^{F,\mathcal{D}} = Accept] = 1$*

  2. *There exists a non-decreasing function $\rho : (0, 1] \to (0, 1]$ (called the detection probability) such that if $F$ is $\varepsilon$-Faraway from $\mathcal{P}$,*
     $$\Pr[\mathcal{T}^{F,\mathcal{D}} = Reject] \geq \rho(\varepsilon).$$

*Here, $\mathcal{T}^{F,\mathcal{D}}$ denotes the execution of the tester $\mathcal{T}$ when given oracle access to the function $F$ and the distribution $\mathcal{D}$.*

To obtain a one-sided error tester with parameter $\varepsilon$, we can make $\Theta\left(\frac{1}{\rho(\varepsilon)}\right)$ independent calls to the POT $\mathcal{T}$ and accept if and only if all the calls accept (Goldreich and Ron, 2008). We denote by $\mathcal{T}^{F,\mathcal{D}}(X)$ the output when applied to a specific sample $X \sim \mathcal{D}$, and note that with abuse of notation we will late consider $\mathcal{D}$ to be the empirical distribution of calibration dataset $\{(X_i, Y_i)\}_{i=1}^n$ in which case we write $\mathcal{T}^{F,\mathcal{D}}(X_i, Y_i)$ for the output on this specific sample from $\mathcal{D}$.

**Example.** Consider functions $f : \mathbb{R}^d \to \mathbb{R}$, and let $\mathcal{P}$ denote the property that $f$ is constant in the $k$-th dimension. This property has has connections to fairness among other applications Caton and Haas (2024). Assume $\mathcal{D}$ is the empirical distribution of the inputs $X \in \mathbb{R}$ for some fixed dataset.

Restrict to set-valued functions $F$ that output compact and connected intervals of the form $[a, b] \subseteq \mathbb{R}$ for $a, b \in \mathbb{R}$. The candidate POT $\mathcal{T}^{F,\mathcal{D}}$ for whether such a set-valued function $F$ accommodates $\mathcal{P}$ is then as follows: sample $X, X' \sim \mathcal{D}$, If $F(X) \cap F(X') \neq \varnothing$, then Accept; otherwise, Reject. We prove that this satisfies Definition 4 in the extended version of the paper Overman et al. (2024).

## 2.2 Conformal prediction and conformal risk control

Our main tool for achieving alignment from this property perspective is built on conformal prediction and conformal risk control (Vovk et al., 2005; Bates et al., 2021; Angelopoulos et al., 2024). Conformal prediction post-processes the outputs of any model $f$ to create prediction intervals $C(\cdot)$ that ensure certain statistical coverage guarantees. Using a calibration dataset $\{(X_i, Y_i)\}_{i=1}^n$ consisting of ground truth input-output pairs, conformal prediction constructs intervals around the predictions of $f$ such that $\Pr[Y_{n+1} \notin C(X_{n+1})] \leq \alpha$ for a user-specified error rate $\alpha$ on a test point $(X_{n+1}, Y_{n+1})$.

This guarantee is notably distribution-free and holds for any function $f$. The probability is over the randomness in all $n + 1$ points; both the calibration set and the test point. The construction of $C(\cdot)$ depends on both the model $f$ and the draw of the calibration data.

The conformal risk control framework extends conformal prediction to notions of error beyond miscoverage (Angelopoulos et al., 2024). Consider a paramater set $\Lambda \subset \mathbb{R}_{\geq 0}$ that is a bounded subset of the nonnegative reals. Given an exchangeable collection of non-increasing, random loss functions $L_i : \Lambda \to (-\infty, B]$, $i = 1, \ldots, n + 1$, conformal risk control uses the first $n$ loss functions and calibration data $\{(X_i, Y_i)\}_{i=1}^n$ to determine $\hat{\lambda}$ such that

$$\mathbb{E}[L_{n+1}(\hat{\lambda})] \leq \alpha.$$

Consider loss functions of the form $L_i(\lambda) = \ell(C_\lambda(X_i), Y_i)$, where $C_\lambda(X_i)$ is a set of outputs constructed by $f$ and the calibration data. Larger values of $\lambda$ generate more conservative prediction sets $C_\lambda(\cdot)$. Let the risk on the calibration data for a given $\lambda$ be $\hat{R}_n(\lambda) = \frac{1}{n}\sum_{i=1}^n L_i(\lambda)$. For a user-specified risk rate $\alpha$, we let

$$\hat{\lambda} = \inf\left\{\lambda : \frac{n}{n+1}\hat{R}_n(\lambda) + \frac{B}{n+1} \leq \alpha\right\}.$$

This choice of $\hat{\lambda}$ guarantees the desired risk control $\mathbb{E}[L_{n+1}(\hat{\lambda})] \leq \alpha$ (Angelopoulos et al., 2024).

# 3 Conformal property alignment

Our main methodology is to use conformal risk control to create prediction intervals that align with specific properties $\mathcal{P}$. Our approach allows us to post-process the outputs of a pre-trained model $f$ to ensure that within the resulting conformal band, with a given probability, there exists predictions that adhere to desired properties such as monotonicity.

## 3.1 Multi-lambda conformal risk control

We make particular use of the conformal risk control algorithm to allow for a $k$-dimensional vector of tuning parameters $\boldsymbol{\lambda} = (\lambda_1, \lambda_2, \ldots, \lambda_k)$, where larger values of $\boldsymbol{\lambda} \in \boldsymbol{\Lambda} \subset \mathbb{R}^k$ yield more

conservative outputs, where $\boldsymbol{\Lambda} \subset \mathbb{R}_{\geq 0}^k$ is a bounded subset of $\mathbb{R}_{\geq 0}^k$. This works by mapping $\boldsymbol{\lambda}$ to a scalar and then applying standard conformal risk control. We emphasize that this result is not new and follows essentially directly from Angelopoulos et al. (2024). The construction of the output set $F_{\boldsymbol{\lambda}}(X) \subseteq \mathcal{Y}$ depends on the specific application and provides flexibility in how the function $f(X)$ and the parameters $\boldsymbol{\lambda}$ are utilized.

**Definition 5** (Construction of $F_{\boldsymbol{\lambda}}(X)$). *Let $f : \mathcal{X} \to \mathcal{Y}$ be a given function. For each $\boldsymbol{\lambda} \in \mathbb{R}^k$, define the set-valued function $F_{\boldsymbol{\lambda}} : \mathcal{X} \to 2^{\mathcal{Y}}$ such that, for each $X \in \mathcal{X}$, $F_{\boldsymbol{\lambda}}(X)$ is a set of predictions for $X$ constructed from $f$ and $\boldsymbol{\lambda}$. The specific construction of $F_{\boldsymbol{\lambda}}(X)$ should satisfy the following properties:*

1. *When $\boldsymbol{\lambda} = \mathbf{0}$, we have $F_{\mathbf{0}}(X) = \{f(X)\}$.*

2. *For any $\boldsymbol{\lambda}, \boldsymbol{\lambda}' \in \mathbb{R}^k$, if $\boldsymbol{\lambda} \leq \boldsymbol{\lambda}'$ (i.e., $\lambda_i \leq \lambda_i' \ \forall i = 1, 2, \ldots, k$), then $F_{\boldsymbol{\lambda}}(X) \subseteq F_{\boldsymbol{\lambda}'}(X)$.*

This definition ensures that increasing the parameters $\boldsymbol{\lambda}$ leads to larger (more conservative) prediction sets, and that when all parameters are zero, the prediction set reduces to the point prediction given by $f(X)$.

Following the original scalar $\lambda$ setting, we assess $F_{\boldsymbol{\lambda}}$ using non-increasing random loss functions $L_i = \ell(F_{\boldsymbol{\lambda}}(X_i), Y_i) \in (-\infty, B]$ for $B < \infty$. In particular, we consider an exchangeable collection of non-increasing random functions $L_i : \boldsymbol{\Lambda} \to (-\infty, B], i = 1, ..., n+1$, where $\boldsymbol{\Lambda} \subset \mathbb{R}_{\geq 0}^k$ is a bounded subset of $\mathbb{R}_{\geq 0}^k$, with bound $\lambda_j^{\max}$ in each dimension $j \in [k]$.

As in Angelopoulos et al. (2024), we use the first $n$ functions to determine $\hat{\boldsymbol{\lambda}}$ so that the risk on the $(n+1)$-th function is controlled, specifically so that $\mathbb{E}[L_{n+1}(\hat{\boldsymbol{\lambda}})] \leq \alpha$.

We apply a similar algorithm. Given $\alpha \in (\infty, B)$ and letting $\hat{R}_n(\boldsymbol{\lambda}) = \frac{L_1(\boldsymbol{\lambda}) + \cdots + L_n(\boldsymbol{\lambda})}{n}$, define

$$\boldsymbol{\Lambda}_{\min} = \min\left\{\boldsymbol{\lambda} : \frac{n}{n+1}\hat{R}_n(\boldsymbol{\lambda}) + \frac{B}{n+1} \leq \alpha\right\}$$

to be the set of minimal elements (Boyd and Vandenberghe, 2004) of $\boldsymbol{\Lambda}$ that satisfy the condition $\frac{n}{n+1}\hat{R}_n(\boldsymbol{\lambda}) + \frac{B}{n+1} \leq \alpha$. Let $g : \boldsymbol{\Lambda} \to \mathbb{R}$ be a strictly increasing function such that $L_i(\boldsymbol{\lambda})$ is non-increasing with respect to the level sets defined by $g(\boldsymbol{\lambda})$. Then select $\hat{\boldsymbol{\lambda}} \in \boldsymbol{\Lambda}_{\min}$ to be a minimizer of $g$ over $\boldsymbol{\Lambda}_{\min}$.

We then deploy the resulting set-valued function $F_{\hat{\boldsymbol{\lambda}}}$ on the test point $X_{n+1}$. For this choice of $\hat{\boldsymbol{\lambda}}$, we have a risk control guarantee that mimics the result of Angelopoulos et al. (2024), specifically:

**Proposition 1.** *Assume that $L_i(\boldsymbol{\lambda})$ is non-increasing with respect to the partial ordering of $\boldsymbol{\Lambda}$ inherited from $\mathbb{R}^k$. Additionally, assume that $L_i(\boldsymbol{\lambda})$ is non-increasing with respect to $g(\boldsymbol{\lambda})$ for some strictly increasing function $g : \boldsymbol{\Lambda} \to \mathbb{R}$. Also assume $L_i$ is right-continuous in each dimension, $L_i(\boldsymbol{\lambda}^{\max}) \leq \alpha$, and $\sup_{\boldsymbol{\lambda}} L_i(\boldsymbol{\lambda}) \leq B < \infty$ almost surely. Then*

$$\mathbb{E}[L_{n+1}(\hat{\boldsymbol{\lambda}})] \leq \alpha.$$

The proof is similar to the proof of the guarantee for the conformal risk control algorithm in Angelopoulos et al. (2024) and is deferred to the extended version of the paper Overman et al. (2024)

To provide intuition on $g(\boldsymbol{\lambda})$, we note that for our primary use case we will take $g(\boldsymbol{\lambda}) = \sum_{i=1}^k \lambda_i$. Clearly this function is strictly increasing in $\boldsymbol{\lambda}$ and intuitively it is reasonable to consider loss functions $L_i$ that are non-increasing as the sum of the components of $\boldsymbol{\lambda}$ increases.

## 3.2 Conformal property alignment from proximity oblivious testers

We now demonstrate how to construct a conformal risk control problem using proximity-oblivious testers (POTs) for a given property $\mathcal{P}$. Suppose we are given a pre-trained model $f : \mathcal{X} \to \mathcal{Y}$. We aim to extend the point predictions of $f$ to prediction sets, where the size or conservativeness of the set is parameterized by a parameter $\boldsymbol{\lambda}$. Let $F_{\boldsymbol{\lambda}} : \mathcal{X} \to 2^{\mathcal{Y}}$ denote the set-valued function that outputs, for each $X \in \mathcal{X}$, the set $F_{\boldsymbol{\lambda}}(X) \subseteq \mathcal{Y}$ determined by $f$, $X$, and $\boldsymbol{\lambda}$.

Let $\mathcal{T}^{F,\mathcal{D}}$ be a proximity-oblivious tester for whether a set-valued function $F$ accommodates the property $\mathcal{P}$ as given y Definition 4. We denote the random output of $\mathcal{T}^{F,\mathcal{D}}$ evaluated at $(X, Y) \sim \mathcal{D}$ by $\mathcal{T}^{F,\mathcal{D}}(X, Y)$.

We now define a loss function, generated from $\mathcal{T}^{F,\mathcal{D}}$, which will be crucial in formulating our conformal risk control problem.

**Definition 6** (Loss Function Generated from a POT). *Let $\mathcal{T}^{F,\mathcal{D}}$ be a proximity-oblivious tester for a property $\mathcal{P}$. We define the loss function $L_i$ as:*

$$L_i = \begin{cases} 0, & \text{if } \mathcal{T}^{F,\mathcal{D}}(X_i, Y_i) = \text{Accept}, \\ 1, & \text{otherwise}, \end{cases}$$

*where $(X_i, Y_i)$ are samples from the distribution $\mathcal{D}$.*

**Example.** Consider the POT for the property $\mathcal{P}$ of a function $f : \mathbb{R} \to \mathbb{R}$ being constant, as mentioned in Section 2.1. Assume we have access to a calibration set $\{(X_i, Y_i)\}_{i=1}^{n}$ of size $n$. We use a two-dimensional parameter $\boldsymbol{\lambda} = (\lambda^-, \lambda^+)$, and define the set-valued function:

$$F_{\boldsymbol{\lambda}}(X) = [f(X) - \lambda^-, \ f(X) + \lambda^+],$$

for each $X \in \mathbb{R}$. This creates prediction intervals around the point prediction $f(X)$, with widths controlled by $\lambda^-$ and $\lambda^+$.

We then apply the loss function generated by $\mathcal{T}^{F_{\boldsymbol{\lambda}},\mathcal{D}}$ as given in Definition 6, and use conformal risk control to tune $\boldsymbol{\lambda}$ such that the expected loss on the $(n+1)$th point falls below a given target level $\alpha$.

Note that in this case the tester and loss function does not depend on the $Y_i$. This is because the property of $f$ being constant does not depend on the $Y_i$ from the calibration set and here $\mathcal{D}$ is only used to obtain samples of the $X_i$. This is not the case in general, however, and properties can be defined with respect to the whole sample $(X_i, Y_i) \sim \mathcal{D}$. For example, we could consider the property $\mathcal{P}$ that $f$ does not over-predict, that is, for $(X, Y) \sim \mathcal{D}$ we have $f(X) \leq Y$. Now we state our main theorem.

**Theorem 1.** *Let $\mathcal{T}$ be a proximity-oblivious tester for a property $\mathcal{P}$ with detection probability function $\rho(\cdot)$. Assume access to a calibration dataset $\{(X_i, Y_i)\}_{i=1}^{n}$ sampled independently from a distribution $\mathcal{D}$. Suppose we run conformal risk control on this calibration dataset using risk parameter $\alpha$ and loss functions $L_i$ for property $\mathcal{P}$ generated from $\mathcal{T}$ (as in Definition 6). Then, for any $\varepsilon$ such that $\rho(\varepsilon) > \alpha$, the probability that $F_{\hat{\boldsymbol{\lambda}}}$ is $\varepsilon$-Faraway from $\mathcal{P}$ satisfies:*

$$\Pr_{(X_1, Y_1), \ldots, (X_n, Y_n)} \left( F_{\hat{\boldsymbol{\lambda}}} \text{ is } \varepsilon\text{-Faraway from } \mathcal{P} \right) \leq \frac{\alpha}{\rho(\varepsilon)}.$$

*Proof.* Let $\mathcal{E}$ denote the event that $F_{\hat{\boldsymbol{\lambda}}}$ is $\varepsilon$-Faraway from the property $\mathcal{P}$. Our goal is to bound the probability $\Pr_{(X_1, Y_1), \ldots, (X_n, Y_n)}[\mathcal{E}]$.

The conformal risk control procedure ensures that the expected loss on a new sample $(X_{n+1}, Y_{n+1})$ satisfies:

$$\mathbb{E}_{(X_1, Y_1), \ldots, (X_n, Y_n), (X_{n+1}, Y_{n+1})}[L_{n+1}] \leq \alpha.$$

Now, we can write

$$\mathbb{E}[L_{n+1}] = \Pr(\mathcal{E}) \cdot \mathbb{E}[L_{n+1} \mid \mathcal{E}] + \Pr(\mathcal{E}^c) \cdot \mathbb{E}[L_{n+1} \mid \mathcal{E}^c].$$

When $\mathcal{E}$ occurs, $F_{\hat{\boldsymbol{\lambda}}}$ is $\varepsilon$-Faraway from $\mathcal{P}$. By the properties of the proximity-oblivious tester $\mathcal{T}$, we have:

$$\Pr_{(X,Y) \sim \mathcal{D}} \left[ \mathcal{T}^{F_{\hat{\boldsymbol{\lambda}}},\mathcal{D}}(X, Y) = \text{Reject} \mid \mathcal{E} \right] \geq \rho(\varepsilon).$$

Thus, the conditional expected loss satisfies:

$$\mathbb{E}[L_{n+1} \mid \mathcal{E}] = \Pr \left[ \mathcal{T}^{F_{\hat{\boldsymbol{\lambda}}},\mathcal{D}}(X_{n+1}, Y_{n+1}) = \text{Reject} \mid \mathcal{E} \right] \geq \rho(\varepsilon).$$

And when since $\Pr(\mathcal{E}^c) \cdot \mathbb{E}[L_{n+1} \mid \mathcal{E}^c]$ is non-negative because $L_{n+1}$ is non-negative, we obtain

$$\mathbb{E}[L_{n+1}] \geq \Pr(\mathcal{E}) \cdot \rho(\varepsilon).$$

Combining this result with our guarantee from the conformal risk control procedure,

$$\alpha \geq \mathbb{E}[L_{n+1}] \geq \Pr(\mathcal{E}) \cdot \rho(\varepsilon).$$

This implies:

$$\Pr(\mathcal{E}) \leq \frac{\alpha}{\rho(\varepsilon)}.$$

Therefore, the probability that $F_{\hat{\boldsymbol{\lambda}}}$ is $\varepsilon$-Faraway from $\mathcal{P}$ satisfies:

$$\Pr_{(X_1,Y_1),\ldots,(X_n,Y_n)} \left( F_{\hat{\boldsymbol{\lambda}}} \text{ is } \varepsilon\text{-Faraway from } \mathcal{P} \right) \leq \frac{\alpha}{\rho(\varepsilon)}.$$

$\square$

**Amplifying Detection Probability via Independent Calls.** When the detection probability $\rho(\varepsilon)$ of the proximity-oblivious tester $\mathcal{T}$ is less than or close to the risk parameter $\alpha$, the bound provided by Theorem 1 may not be tight or meaningful (since $\alpha/\rho(\varepsilon)$ could be greater than or equal to 1). To address this issue, we can amplify the detection probability by performing multiple independent executions of $\mathcal{T}$ and combining their results appropriately.

To increase the detection probability beyond $\alpha$, we execute the proximity-oblivious tester $\mathcal{T}$ independently $k$ times on independent samples and define a new tester $\mathcal{T}'$ that rejects if *any* of the $k$ executions reject (i.e., by applying a logical OR to the outcomes). This amplification technique yields an adjusted detection probability

$$\rho'(\varepsilon) = 1 - (1 - \rho(\varepsilon))^k,$$

representing the probability that at least one of the $k$ independent executions rejects when the function is $\varepsilon$-Faraway from $\mathcal{P}$.

In this approach, the calibration dataset needs to be partitioned into $n' = \lfloor \frac{n}{k} \rfloor$ disjoint batches, each containing $k$ samples. Each batch provides the independent samples required for the $k$ executions of $\mathcal{T}$ per calibration point. As a result, the effective sample size available for calibrartion becomes $n'$ due to this batching of samples.

## 4 Examples

### 4.1 Monotonicity

Monotonic behavior is important in various applications. We focus on monotonicity in a single feature, where we expect that $f(X)$ should have monotonically increasing or decreasing behavior with respect to a certain feature $x^k$ when other features $x^{-k}$ are held fixed. While there is a long-standing literature on using monotonic constraints for regularization (Brunk et al., 1973; Sill and Abu-Mostafa, 1996; You et al., 2017; Bonakdarpour et al., 2018) and on integrating such monotonic shape constraints into prediction models (Groeneboom and Jongbloed, 2014; Cano et al., 2018; Runje and Shankaranarayana, 2023), our aim is not to view monotonicity as a possible means to improve test accuracy, but rather as a user-desired property for safe or fair deployment of a given model. For example, Wang and Gupta (2020) highlight the importance of monotonicity in models for criminal sentencing, wages, and medical triage.

Consider a user given a pre-trained model $f$ that was not trained with monotonic constraints. The user, however, wishes for the sake of safe or fair deployment to make predictions in a way that is as monotonic as possible. In particular, let $\mathcal{P}$ be the property that $f$ is monotonically decreasing in dimension $k$. To apply our methodology we consider the proximity oblivious tester $\mathcal{T}$ for $\mathcal{P}$ as given in Algorithm 1.

We prove in the extended version of the paper Overman et al. (2024) that Algorithm 1 is indeed a POT for the property $\mathcal{P}$ of being monotonically decreasing in a given dimension. Then let $\mathcal{M}$ be the one-sided error tester for $\mathcal{P}$ resulting from $\Theta(1/\rho(\varepsilon))$ calls to $\mathcal{T}$. Now assume we have access to a calibration dataset $\{(X_i, Y_i)\}_{i=1}^n$ sampled from $\mathcal{D}$ of size $n \in \Omega(1/\rho(\varepsilon))$. We will use this calibration dataset to determine the setting of $\boldsymbol{\lambda} = (\lambda^+, \lambda^-)$ via conformal risk control where the loss function is generated as in Definition 6. Here the set-valued function will be constructed as $F_{\boldsymbol{\lambda}}(X) = [f(X) - \lambda^-, f(X) + \lambda^+]$. Then by Theorem 1 if the tester has sufficient detection probability $\rho(\varepsilon) > \alpha$ we expect to obtain a set-valued function $F_{\hat{\boldsymbol{\lambda}}}$ at most $\varepsilon$ from $\mathcal{P}$. We now investigate this empirically.

**Algorithm 1** POT $\mathcal{T}$ for property $\mathcal{P}$ of monotonically decreasing in dimension $k$

---
1: Sample $X_1 \sim \mathcal{D}$. Let $X_1 = (x_1, x^{-k})$
2: Sample $x_2$ from the marginal distribution of $\mathcal{D}$ in dimension $k$. Set $X_2 = (x_2, x^{-k})$
3: **if** $x_1 < x_2$ and $\max F(X_1) < \min F(X_2)$ **then**
4:     **return** Reject
5: **else if** $x_2 < x_1$ and $\max F(X_2) < \min F(X_1)$ **then**
6:     **return** Reject
7: **end if**
8: **return** Accept

---

**Setup.** We align for monotonicity on various UCI ML repository datasets (Dua and Graff, 2023) with a 70-15-15 train-calibrate-test split, averaged over 30 random splits. We use XGBoost regression models (Chen and Guestrin, 2016). For each dataset, we select a feature for which we desire the model to be monotonic, not with the intention of improving test-set accuracy, but from the perspective of a user who desires this property.

We train two models per dataset: one unconstrained, trained on the training set, and another constrained to be monotonic, trained on both the training and calibration sets. The conformal risk control procedure is applied to the unconstrained model using the calibration data. The constrained model can be considered best possible from the user's perspective, using all available pre-test data and satisfying the monotonicity property $\mathcal{P}$ during training.

To compare performance with respect to the training metric of accuracy, we convert conformal intervals into point predictions by taking $k$-quantiles of the constrained feature, linearly interpolating between adding $\lambda^+$ at the lowest quantile to subtracting $\lambda^-$ at the highest quantile for monotonically decreasing, or vice versa for monotonically increasing.

**Results.** Table 4.1 presents results on the test set for the Combined Cycle Power Plant dataset (Tfekci and Kaya, 2014). In practice, Exhaust-vacuum is known to negatively influence turbine efficiency (Tfekci and Kaya, 2014). The conformal procedure outperforms the constrained model in terms of MSE for all $\alpha$, which is a fortuitous but unexpected outcome. The constrained model should be seen as an oracle benchmark in the sense that the model was given to the user already trained to satisfy the desired property. The risk metric closely matches the theoretical guarantee from conformal risk control and achieves optimal performance of 0 for the constrained model. Additional datasets and results are detailed in the extended version of the paper Overman et al. (2024).

Table 1: Power Plant, $n = 9568$. Monotonically decreasing on Exhaust Vacuum. $\boldsymbol{\lambda}^{\mathrm{max}} = (10, 10)$.

| $\alpha$ | $\boldsymbol{\lambda}$ | Metric | Unconstrained | Adjusted | Constrained |
|---|---|---|---|---|---|
| 0.1 | $\lambda^+ = 0.51_{(\pm 0.24)}$ $\lambda^- = 0.76_{(\pm 0.24)}$ | MSE Risk | $10.19_{(\pm 0.46)}$ $0.75_{(\pm 0.09)}$ | $10.47_{(\pm 0.46)}$ $0.10_{(\pm 0.001)}$ | $16.21_{(\pm 0.45)}$ $0.00_{(\pm 0.00)}$ |
| 0.05 | $\lambda^+ = 1.09_{(\pm 0.51)}$ $\lambda^- = 1.61_{(\pm 0.50)}$ | MSE Risk | $10.19_{(\pm 0.46)}$ $0.75_{(\pm 0.09)}$ | $11.42_{(\pm 0.44)}$ $0.05_{(\pm 0.001)}$ | $16.21_{(\pm 0.45)}$ $0.00_{(\pm 0.00)}$ |
| 0.01 | $\lambda^+ = 2.39_{(\pm 0.82)}$ $\lambda^- = 3.33_{(\pm 0.79)}$ | MSE Risk | $10.19_{(\pm 0.46)}$ $0.75_{(\pm 0.09)}$ | $14.46_{(\pm 0.48)}$ $0.01_{(\pm 0.001)}$ | $16.21_{(\pm 0.45)}$ $0.00_{(\pm 0.00)}$ |

## 4.2 Concavity

Concavity and convexity are crucial behaviors in many applications. In this context, we focus on concavity in a single feature. A common example where users might expect concave behavior is in recommendation or preference prediction models. According to economic theory, the utility function with respect to the quantity of an item is often quasi-concave, reflecting the principle of diminishing marginal utility (Mas-Colell et al., 1995). Jenkins et al. (2021) propose a novel loss function to account for this expected concavity, which aligns the model with the concavity property $\mathcal{P}$ during training. Here we again consider aligning a pre-trained model, not trained to satisfy $\mathcal{P}$, using a proximity oblivious tester $\mathcal{T}$ for $\mathcal{P}$ as described in Algorithm 2.

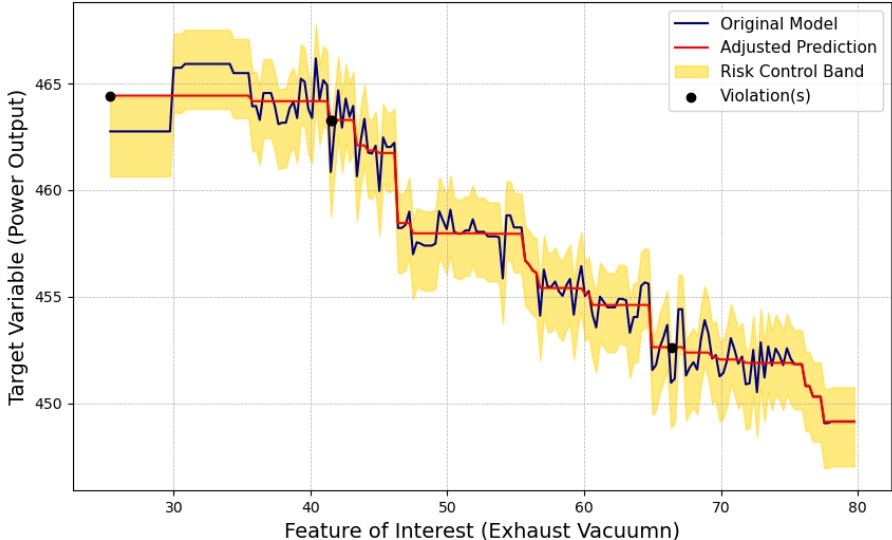

Figure 1: Univariate partial dependence plot of unconstrained model. Risk control band for $\alpha = 0.05$. Dashed line exemplifying Theorem 1 demonstrating existence of monotonically decreasing function falling within the conformal band on $0.975 > 1 - \alpha$ fraction of the domain.

---

**Algorithm 2** POT $\mathcal{T}$ for property $\mathcal{P}$ of concavity in dimension $k$

---

1: Sample $X_{\text{mid}} \sim \mathcal{D}$
2: Sample $\delta_{\text{left}}, \delta_{\text{right}}$ from empirical differences in feature $k$
3: Set $X_{\text{left}}$ by decreasing feature $k$ of $X_{\text{mid}}$ by $\delta_{\text{left}}$
4: Set $X_{\text{right}}$ by increasing feature $k$ of $X_{\text{mid}}$ by $\delta_{\text{right}}$
5: Query $F(X_{\text{mid}})$, $F(X_{\text{left}})$, and $F(X_{\text{right}})$
6: Compute $\alpha = \dfrac{X_{\text{right}}[k] - X_{\text{mid}}[k]}{X_{\text{right}}[k] - X_{\text{left}}[k]}$
7: **if** $\min F(X_{\text{mid}}) > \alpha \max F(X_{\text{left}}) + (1 - \alpha) \max F(X_{\text{right}})$ **then**
8:     **return Reject**
9: **end if**
10: **return Accept**

---

We can use a calibration dataset to determine the setting of $\boldsymbol{\lambda} = (\lambda^+, \lambda^-)$ via conformal risk control where the loss function is generated as in Definition 6. The set-valued function will be constructed as $F_{\boldsymbol{\lambda}}(X) = [f(X) - \lambda^-, f(X) + \lambda^+]$. We demonstrate running conformal risk control with this loss function on a real-world dataset in the extended version of the paper Overman et al. (2024)

## 5  A stylized examination of alignment persistence in AI models

Consider data generated as:
$$y = g(X) + h(X) + \varepsilon\,,$$
where $\varepsilon$ is mean-zero noise with variance $\tau^2$ independent of $X$. Here, $h(X)$ is biased noise we want to ignore, aiming to learn only $g(X)$. Consider the case in which experts expect data to follow $g(X) + \textit{unbiased noise}$, but biased noise $h(X)$ can obscure this.

One potential reason for the presence of biased noise in data could be due to a measurement error of the outcome that is correlated with select features, leading to an incorrectly calculated outcome. A biased measurement error could occur if there is incomplete data and the presence of the incomplete

data is correlated with select features in an unexpected, systematic way. Our goal is to understand how this bias affects model behavior when trying to learn $g(X)$ alone.

Given $n$ i.i.d. samples $\{(X_i, Y_i)\}_{i=1}^n$ from the above model, we denote this dataset by $\mathcal{D}_n$. We use a random feature model:

$$f_{\mathsf{RF}}(X; \mathbf{a}, \{\mathbf{w}_j\}_{j \in [N]}) = \frac{1}{\sqrt{N}} \sum_{j \in [N]} a_j \sigma(\langle X, \mathbf{w}_j \rangle),$$

where $\mathbf{a} \in \mathbb{R}^N$ are learned weights, and $\{\mathbf{w}_j\}_{j \in [N]}$ are fixed random weights. The squared loss is minimized by ridge regression:

$$\hat{\mathbf{a}}_\lambda = \arg \min_{\mathbf{a} \in \mathbb{R}^N} \sum_{i \in [n]} (Y_i - f_{\mathsf{RF}}(X_i))^2 + \lambda \|\mathbf{a}\|_2^2.$$

Users expect a model to exhibit a property $\mathcal{P}$, satisfied by $g(X)$ but not necessarily by $g(X) + h(X)$. We can constrain training to ensure $\mathcal{P}$. Let $C_\mathcal{P} = \{\mathbf{a} \mid \mathbf{a} \in \mathbb{R}^N \text{ and } f_{\mathsf{RF}}(X; \mathbf{a}) \text{ satisfies } \mathcal{P}\}$, yielding a constrained model: $\hat{\mathbf{a}}_{\lambda, \mathcal{P}} = \arg \min_{\mathbf{a} \in C_\mathcal{P}} \sum_{i \in [n]} (Y_i - f_{\mathsf{RF}}(X_i))^2 + \lambda \|\mathbf{a}\|_2^2$.

Assuming $g$ and $h$ are polynomials with $\deg_g < \deg_h$, and given specific conditions on data size and model parameters, we consider two settings: (i) *Classic:* $d^{\deg_g + \delta} < N < d^{\deg_h - \delta}$, and (ii) *Underspecified:* $N > d^{\deg_h + \delta}$ for a small $\delta > 0$.

In the extended version of the paper Overman et al. (2024), we utilize results from Misiakiewicz and Montanari (2023) to derive insights into the impact of model complexity and data size on adherence to $\mathcal{P}$. In particular, we show that under certain assumptions, including small noise bias and robustness of property $\mathcal{P}$, the constrained and unconstrained models have zero distance in the classic setting: $\hat{\mathbf{a}}_{\lambda, \mathcal{P}} = \hat{\mathbf{a}}_\lambda$. However, in the underspecified setting, the constrained and unconstrained models will differ, resulting in a non-zero distance: $\hat{\mathbf{a}}_{\lambda, \mathcal{P}} \neq \hat{\mathbf{a}}_\lambda$. This result implies that in the presence of noise bias, the overparameterized models (i.e., underspecified setting) fail to satisfy the property $\mathcal{P}$, and this cannot be remedied as the data size increases.

# 6 Related work

Our paper draws from a broad range of areas, hence we refer the reader to textbooks and surveys in alignment (Everitt et al., 2018; Hendrycks et al., 2022; Ji et al., 2024; Hendrycks, 2024), conformal prediction (Angelopoulos and Bates, 2022), property testing (Ron, 2008; Goldreich, 2017), and linearized neural networks (Misiakiewicz and Montanari, 2023).

RLHF (Christiano et al., 2017) has been notably effective in aligning LLMs with human values and intentions, as demonstrated by (Ouyang et al., 2022). Our work considers attempts at alignment that generalzies to models without human-interpretable outputs, which has connections to the scalable oversight problem (Irving et al., 2018; Christiano et al., 2018; Wu et al., 2021). Goal misgeneralization (Langosco et al., 2022; Shah et al., 2022) has potential connections to the underspecified pipeline (D'Amour et al., 2022) considered in this paper in the sense that models with equivalent performance according to the training metric may differ in some other user-desired property during deployment. One of the main methods of assurance (Batarseh et al., 2021), which is concerned with assessing the alignment of pre-trained AI systems, is safety evaluations (Perez et al., 2022; Shevlane et al., 2023) meant to assess risk during deployment, which also has connections to our approach.

The work of Yadkori et al. (2024) closely aligns with ours in both methodology and theme, utilizing conformal risk control to reduce LLM hallucinations (Ji et al., 2023). We discuss connections to this work in the extended version of the paper Overman et al. (2024).

# 7 Discussion

We introduce a method to align pre-trained models with desired user properties using conformal risk control. By post-processing outputs using property dependent loss functions, we provide probabilistic guarantees that conformal intervals contain functions close to the desired set. This allows for alignment without retraining, effective in both generative and non-generative contexts. Future work should extend these techniques to more properties, explore sample complexity and adaptive querying, and potentially apply them to policy functions in MDP settings for RL agent safety guarantees.

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
