# OpenReview forum: "Aligning Model Properties via Conformal Risk Control"
_NeurIPS.cc/2024/Conference — NeurIPS 2024 poster_

### Official Review · Reviewer_Hsyg · 2024-07-10

**Soundness:** 3
**Presentation:** 4
**Contribution:** 3
**Rating:** 7
**Confidence:** 4

**Summary:**

The paper connects conformal risk control to define prediction sets containing results that satisfy a property. In short, assume that function $f$ (a trained model) does not satisfy some property $\mathcal{P}$, we can define a prediction interval around results of $f$ s.t. those intervals satisfy $\mathcal{P}$ with $1 - \alpha$ probability. The authors first generalize conformal risk control to multidimensional conservative parameter, through a proxy from $\mathbb{R} ^d \mapsto \mathbb{R}$, and use that in addressing the property testing. They define the risk as if there is no output in the prediction interval that satisfies the property.

**Strengths:**

The idea is novel. Defining conformal property satisfaction based on property testing and conformal risk control opens a new area of work based on black-box post-hoc modifications of the model toward a desired property. I believe it can be further used in model-explainability, fairness, or tweaking the model toward constrained predictions.

The idea is nicely developed and nicely presented. It can easily engage an audience how is not familiar with conformal risk control or property testing. I would recommend an additional introduction to the aforementioned topics in the appendix, however the current version is also really nice.

I also find the way the authors address fairness in section 5 (through bias in measurement error) really interesting.

**Weaknesses:**

1. Notations in the paper can be introduced in a better way. For instance a mathematic notation of $\mathcal{P}$, which can be a subset of the function space.

2. There are minor typos in the mathematical notation e.g. in line 88, $x \sim \mathcal{D}$ but $f(X) \neq g(X)$. Also I do not understand why the authors used $\mathrm{Min}$ instead of $\min$; I believe that might have to relate to the fact that they are minimizing over vectors but still it could be indicated by $\min_{\boldsymbol{\lambda}\in\boldsymbol{\Lambda}}$. Also shouldn't we find the minimal $\boldsymbol{\lambda} _\mathrm{min}$ instead of $\boldsymbol{\Lambda} _\mathrm{min}$?

**Questions:**

1. Why did the authors choose monotonicity and concavity as the properties to test. Is there any real-life use-case of those properties or does more justifiable properties remain as a possible future work? Ofcourse there is a dicussion on applications of monotonicity to causal inference, and fairness but is there any chance that the authors could provide experiments leading to those areas?

2. Definition 2 has some confusing points:
- Should there be any nested property over the prediction sets for various lambdas? Like if $\lambda_a \le \lambda_b$ then $\mathcal{C}(x_i; \lambda_a) \subseteq \mathcal{C}(x_i; \lambda_b)$? If yes then how the order is defined in this vector space? Does the nested property hold for a linear combination of $\lambda_i$ values or the nested property should be present for any $\lambda_i$.
- How the interval is defined over function? Is it just the function value $\pm$ some radius around it or is it a something apply over function parameters?

3. In line 156, shouldn't we find the minimal $\boldsymbol{\lambda}_\mathrm{min}$ instead of $\boldsymbol{\Lambda}_\mathrm{min}$?

**Limitations:**

To the best of my understanding, the paper defines the guarantee over a property, and a joint guarantee of property + accuracy is not defined in its framework. I see that their extension of risk control supports multi-dimensional set constant not a multi-dimensional risk. However, I believe this is not a shortcoming of the method but a possible open problem to solve.

---

> ### Author Rebuttal · Authors · 2024-08-06
>
> We would like to thank the reviewer for a positive and confident assessment of our paper and appreciate the clear attention to detail shown by the reviewer. Both the strengths and weaknesses/questions/limitations make it evident to us that the reviewer invested significant time understanding the core of our paper.
>
> We thank the reviewer for the direct assessment of our work as novel and capable of opening a new area in post-hoc modification of models to align with desired properties. We also thank the reviewer for finding our paper engaging to read and potentially applicable to domains such as explainability and fairness, and appreciate the interest given to Section 5 on the random feature model theory and its connections to alignment and fairness.
>
> We find all of the reviewer's listed weaknesses helpful and easily addressable. We completely agree with the reviewer on the listed notation fixes and typos, we are happy to address those in a final version. In particular, we agree that we can use a lower case $\min$ notation in place of Min. With regards to the question of finding the minimal $\lambda_{\text{min}}$ instead of $\Lambda_{\text{min}}$, this distinction was drawn to clarify that the set of minimal elements (which can have cardinality more than 1) is first formed by finding the set of $\lambda$ that give a bounded risk below $\alpha$, and then $\lambda_{\text{min}}$ is found by finding an argmin of the function $g$ from this initial set. Ultimately we agree that the notation in this section should be more clearly clarified to avoid confusion. Thanks to the reviewer for highlighting this and delving into detail on the multi-lambda risk control algorithm.
>
> We thank the reviewer for their questions that clearly indicate engagement with our work.
> For question 1, we feel that monotonicity is a natural first property to study in this framework. Monotonicity is well-established as a desirable property in many settings as described by the Wang and Gupta (2020) paper we cite at the beginning of Section 4.1. Moreover, it is also well-studied in the property testing literature. Following the reviewer's request for additional experiments, we next demonstrate new experiments in fairness and medicine to highlight the usefulness of monotonicity.
>
> Although we lack the time or scope to provide a comprehensive comparison of our method to the extensive fairness literature, given the reviewer’s request we have included in the pdf document attached to the global rebuttal a figure for results of applying our method to the UCI ML repo “Student Performance” dataset. This dataset is often used as a benchmark in the fairness literature (see “A survey on datasets for fairness-aware machine learning” Quy et al. 2022). Following the literature, we take “sex” to be the protected attribute and the final grade “G3” to be the real-valued outcome of interest. The attached figure demonstrates the lambdas needed to achieve a constant predicted grade when varying “sex” with a tolerance of 0.1 for $\alpha$.
>
> Also given the reference made to applications of monotonicity in medical triage at the beginning of Section 4.1, we have included another figure in the attached pdf showing the results of our method on the UCI ML repo “Liver Disorders” dataset. Here we take the target of average number of drinks per day (specified as target in the repo) to be monotonically increasing in the feature “mcv,” of which increasing levels are known to be predictive of liver disorders. This example from the high-stakles medical domain emphasizes that we believe monotonicity has significant importance in improving trust, interpretability, and hence adoption of clinical models.
>
> For question 2, we wish to thank the reviewer for sharing their confusion with Definition 2. This definition was intentionally abstract, but we realize now it may have lacked sufficient clarity.
>
> In the original CRC paper, $C_\lambda$ is just taken to be any arbitrary function of the model and calibration data that outputs a prediction set, and outputs an increasingly large set as $\lambda$ increases. It is not stated explicitly but appears to be assumed implicitly that if $\lambda_a \leq \lambda_b$, then $C_{\lambda_a} (X_i) \subseteq C_{\lambda_b} (X_i)$.
>
> We operate under the same implicit assumption that for $\lambda_a, \lambda_b \in \mathbb{R}^k$, if $\lambda_a \leq \lambda_b$, then $C_{\lambda_a} \subseteq C_{\lambda_b}$. However, this does not have to be strictly increasing; we can have $\lambda_a < \lambda_b$, but $C_{\lambda_a} = C_{\lambda_b}$. This nested property holds for every dimension due to the fact that we follow the partial ordering of $\mathbb{R}^k$ as defined in the main text.
>
>
>
> For the question of how the interval is defined over function, we can define this as $C_{\boldsymbol{\lambda}}(f):\mathcal{X} \to \mathcal{P}(\mathcal{Y})$ is a set valued function where at each point $X_i$ we have $C_{\boldsymbol{\lambda}}(f)(X_i) = C_{\boldsymbol{\lambda}}(X_i)$. Thus it does not necessarily have to be a radius around the function output $f(X_i)$, but this is natural when operating in the reals. We point the reviewer to our discussion of the connection to Yadkori et al. (2024) within Section 6 Related work, which exemplifies a use case of our methodology that does not involve real-valued outputs. We hope the above discussion helps make Definition 2 more mathematically precise, and we look forward to the chance to revise this definition in a final version.
>
>
> For question 3, similar to the discussion above on the lambda notation we agree that this can be improved and will do so for the final version.
>
> Finally, we agree with the reviewer with respect to both of the two stated limitations and also believe that these are exciting avenues for future research.
>
> Again we would like to thank the reviewer for their compellingly positive and extensive assessment of our work.

---

### Official Review · Reviewer_8oYs · 2024-07-12

**Soundness:** 3
**Presentation:** 3
**Contribution:** 3
**Rating:** 5
**Confidence:** 4

**Summary:**

This article offers an alternative approach to alignment by testing whether trained models follow specific properties. This is done, for instance, on monotone or concave functions. A small modification for the use case is made to the Conformal Risk Control setting to allow vectorial parameters. Using CRC and property testers (POTs), a guarantee is obtained that the predictor approximately satisfies the property.

**Strengths:**

The topic of the paper is interesting, and the approach is novel, to my knowledge.
This approach's applications are quite broad and should interest communities interested in alignment, robustness, or, for instance, linking to certain physical applications. A few applications are presented in experiments.
Finally, I find the paper, overall, well written.

**Weaknesses:**

However, several parts could be improved. Section 2.1 is unclear, and the introduction of lambda without details is bound to confuse readers not familiar with CRC.
Moreover, I found section 3.2 to be overly verbose and lacking clear, explicit mathematical formulations of the loss, particularly but not exclusively.
In case it is a misunderstanding, I believe a rewriting and clear formalism would help in understanding.
Moreover, I want to emphasize the lack of links between different sections of the papers. Notations, like the function "g" are reused with different meanings, and the threshold "epsilon" disappears from the method.

Additional typo: line 147 "setof".

**Questions:**

Although early in the article, it's specified that " we can imagine a user who wishes to determine whether a pre-trained model f belongs to P but is unable to train a model of f ’s size" (42), line 178 proposes to "modify" the model. How is that done?
Moreover, in the appendix (table 4), unconstrained models have varying performance when alpha varies, how is that explained? (moreover, no notion of uncertainty is provided on the metrics)

**Limitations:**

I believe it's worth mentioning that this multidimensional  CRC is essentially one-dimensional, as it replaces the non-increasingness in lambda by one in a mapping of lambda to R.
Moreover, the monotonicity in the l1 norm of lambda is often not verified (think of different slopes for the different dimensions), and several experiments do not use this result.

---

> ### Author Rebuttal · Authors · 2024-08-06
>
> We would like to thank the reviewer for what we believe is an overall positive assessment of our paper. We especially appreciate that they find the topic interesting and our approach novel. We also agree this is a standout strength of this work. Additionally we thank the reviewer for observing that the generality and flexibility of our approach can make the work attractive to a broad audience, including alignment researchers, AI safety in robotics and other physical applications, etc.
>
> We also feel optimistic about our ability to alleviate the reviewer’s questions, perceived weaknesses, and remarks concerning limitations of this work. We find almost all of the comments to be simultaneously fair and readily addressable.
>
> We begin with the questions. Firstly, we address the question “Although early in the article, it's specified that, ‘we can imagine a user who wishes to determine whether a pre-trained model $f$ belongs to $\mathcal{P}$ but is unable to train a model of $f$ ’s size’ (42), line 178 proposes to "modify" the model. How is that done?”
>
> The “modification” to f that we are referring to in Definition 3 is abstract in the sense that it is considering the case in which $f$ had output $\tilde{Y}$ instead of its actual output $f(X)$. No actual modification to the model is made, the underlying function $f$ remains a black box. We will edit this definition to make this point more clear. We thank the reviewer for bringing this confusion to our attention.
>
> The second question, “Moreover, in the appendix (table 4), unconstrained models have varying performance when alpha varies, how is that explained? (moreover, no notion of uncertainty is provided on the metrics)” is even easier for us to address. This was simply a typo; the 114 for $\alpha=0.05$ should have been 115 as for $\alpha=0.1$ and $\alpha=0.01$, after which you can observe the performance of the unconstrained model is consistent across varying $\alpha$. We thank the reviewer for pointing out this typo.
>
> The reviewer expressed concerns regarding multi-lambda risk control. We agree that the generalization is rather straightforward from the original conformal risk control and essentially mimics the one-dimensional case due to the mapping to $R$. We attempt to clearly state in both the proposition statement in the main text and the proof in the appendix that the proof closely mimics the original result. It is not our intention to frame this result as a significant technical contribution but rather a novel perspective on conformal risk control with application to alignment.
>
> We also believe that even if the generalization to multi-lambda conformal risk control is straightforward, the literature has overlooked it so far, even in instances in which it would have been helpful. In particular, as mentioned in the global response, Yadkori et al. (2024) utilize two separate conformal procedures to calibrate two separate functions, which could have been simultaneously calibrated in tandem using multi-lambda conformal risk control. Other works, such as “Conformal Risk Control for Ordinal Classification” (Xu, 2023), assume that lower and upper bounds on a prediction set are given by functions l(lambda) and u(lambda) but requiring both ends of the interval to grow simultaneously may not be generally necessary; multi-lambda conformal risk control provides a flexible implementation of the conformal risk control algorithm in such settings.
>
> We thank the reviewer for informing us that Section 2.1, introducing the preliminaries for property testing, is insufficiently clear. We hope to better explain the main ideas needed from property testing in this section and provide a more comprehensive introduction to property testing in the appendix in the final version, as suggested by Reviewer Hsyg.
>
> We appreciate the feedback on insufficient clarity regarding the definition of the loss function in Section 3.2. We can rewrite this definition similarly to as follows:
>
> $$
> L_i = \ell \left( C_\lambda (X_i), Y_i \right) =
> 0 \text{ if } \exists \hat{Y} \in C_\lambda (X_i) \text{ s.t. } T_i (\hat{f}, X_i, Y_i) = \text{Accept} \text{, otherwise } 1
> $$
>
> $$
> \text{where} \quad \hat{f} (X) =
> f(X) \text{ if } X \not= X_i \text{, or } \hat{Y} \text{ if } X = X_i
> $$,
>
> Where $T_i$ for $i=1,...$ is an infinite sequence of testers with fixed randomness for each $i$. That is, each $T_i$ is a deterministic function of its inputs $(\hat{f}, X_i, Y_i)$.
>
> We hope this definition provides sufficient clarity on the loss function.
>
> We also look forward to the chance to clean and align the notation around $g$ and $\varepsilon$, as mentioned by the reviewer. Thanks to the reviewer for pointing out the typo “setof” on line 147. We will fix these errors in the final version.
>
> We hope all of these responses are satisfactory to the reviewer. As mentioned at the beginning, we appreciate that the reviewer gave us an overall positive review with respect to the significance, novelty, and applicability of our work. We also appreciate the 3s across the board from the reviewer on Soundness, Presentation, and Contribution. We hope our comments above have sufficient gravity to positively shift the reviewer’s overall score and assessment of our paper.

---

> > ### Comment · Reviewer_8oYs · 2024-08-13
> >
> > I appreciate the author's rebuttal, the improved clarity, and the fact that some inconsistencies were typos. I will increase my score accordingly.

---

> > > ### Author Response · Authors · 2024-08-13
> > >
> > > We appreciate the reviewer's increased score. Thank you for the continued engagement with our paper and for appreciation of the improved clarity and corrected typos. Thank you again.

---

### Official Review · Reviewer_7tC7 · 2024-07-13

**Soundness:** 3
**Presentation:** 3
**Contribution:** 3
**Rating:** 7
**Confidence:** 3

**Summary:**

The authors propose a way to solve the problem of alignment in Machine Learning using Conformal Risk Control. They first expand the previous work of conformal risk control to multidimensional parameters $\boldsymbol{\lambda}$, then used this extension to propose a way to test if a function belongs to a certain class of functions $\mathcal{P}$ that we assumed are aligned with user interests. Finally, they demonstrate how this method can be used to test monotonicity of functions.

**Strengths:**

* The theoretical foundations of the paper seems pretty solid. The methodology is clear and the paper is quite easy to follow
* The monotonicity and concavity examples are pretty convincing
* Detailing the linearity case help understand the theoretical foundation and reasoning of the method

**Weaknesses:**

* There is only a single example with results. Presenting a method for concavity is interesting but it would be nice to see it applied to a real world examples
* There is still a significant gap between monotonicy/concavity and alignment as we understand it in AI.

**Questions:**

see weakness in concavity.

**Limitations:**

* The authors advertise this paper as being useful in GenAI. What are examples of potential class of functions that would satisfy alignment with a generated text / image for example. It is indeed a good first step to test the belonging to a class, but defining that class can be very hard in the most important application

---

> ### Author Rebuttal · Authors · 2024-08-06
>
> We would like to thank the reviewer for their positive assessment of our paper. We appreciate the recognition of the solid theoretical foundations, the clarity of our methodology, and the overall readability of the paper. We also value the opportunity to address your concerns and clarify our contributions.
>
> Regarding the reviewer’s first stated weakness: "There is only a single example with results. Presenting a method for concavity is interesting, but it would be nice to see it applied to real-world examples," we believe there may be a misunderstanding. We included two additional examples on real datasets for monotonicity in the appendix and a real-world example for the concavity method. We briefly mentioned these at the end of the Results section in 4.1, but we will make this more explicit in the final version. We hope the reviewer finds these additional examples convincing of the strength of our paper.
>
> The reviewer’s other stated weakness concerns the gap between monotonicity/concavity and alignment in AI. We appreciate the desire for a more detailed discussion on this matter. As discussed in our abstract and introduction, our goal is to propose a framework for interpreting and performing alignment in settings with outputs less interpretable and amenable to human feedback. Thus, our goal was to show that this framework could extend from properties like monotonicity to more complex properties of generative models.
> Our methodology allows us to define and align desirable properties of generative models, provided we can test for these properties at a per-input level. We included this discussion in our general response but will restate it here for the reviewer’s convenience.
>
> We want to highlight this claim via a section that may have been too buried originally. The connection to Yadkori et al. (2024) in Section 6 Related Work was not only to acknowledge their work but also to highlight the applicability of our methodology to a generative AI setting. We discuss that their use of conformal risk control to mitigate LLM hallucination can be captured as a specific case of our more general property alignment perspective.
>
> We discuss that “not hallucinating” can be considered a property of the function $(a, f)$, where $a$ determines whether the model should abstain, and if not, the function $f$ gives the output. This example shows how our approach of forming the conformal intervals is not limited to subsets of the reals since, in this application, the conformal interval is either just $f(X)$ or $\{f(X), Abstain\}$ for sufficiently conservative lambda.
>
> While including an application of our approach for LLM alignment is beyond the scope of this paper and not feasible for this rebuttal, we believe this discussion highlights the potential applicability of our approach to generative AI. Our methodology potentially enables one to define and align desirable properties of generative models as long as there is a way to test for these properties at a per-input level.
> We also wish to draw attention to the importance of monotonicity through an additional example provided in Figure 1 of the attached PDF. Although we lack the time or scope to provide a comprehensive comparison of our method to the extensive fairness literature, we have included a figure for results of applying our method to the UCI ML repo “Student Performance” dataset. This dataset is often used as a benchmark in the fairness literature (see “A survey on datasets for fairness-aware machine learning” Quy et al. 2022). We take “sex” to be the protected attribute and the final grade “G3” to be the real-valued outcome of interest. The attached figure demonstrates the lambdas needed to achieve a constant predicted grade when varying “sex” with a tolerance of 0.1 for $\alpha$.
>
> Given the reference to applications of monotonicity in medical triage at the beginning of Section 4.1, we have included Figure 2 in the attached PDF showing the results of our method on the UCI ML repo “Liver Disorders” dataset. Here, we take the target of the average number of drinks per day (specified as the target in the repo) to be monotonically increasing in the feature “mcv,” which is known to be predictive of liver disorders. This example from the high-stakes medical domain emphasizes that monotonicity significantly improves trust, interpretability, and the adoption of clinical models.
>
> We also point the reviewer to another newly included experiment in Figure 3a and 3b at the end of the general response, highlighting the applicability of the framework to a more complex safety alignment problem of avoiding side-effects. The details of this experiment are included in the global response.
>
> We hope these examples demonstrate that while monotonicity and concavity may not match the usual notion of alignment in generative AI, they are important properties of models. Our methodology allows for application to more complex settings, such as generative AI and RL.

---

> > ### Comment · Reviewer_7tC7 · 2024-08-12
> >
> > Thank you for your taking the time to provide this detailed answer.
> >
> > I was also questioning the improvement from the multidimensional $\lambda$ but have been convinced by your answer to reviewer 9EgD.
> >
> > The applications to LLMs with the non hallucination is also pretty convincing. Running an experiment against Yakdori et al. (2024) method would have made it clearer to the reader on how well it improves from that baseline to use the multidimensional $\lambda$ instead of separate ones, but obviously there is not enough time for it.
> >
> > I still  think 7 is the appropriate rating for this paper.

---

> > > ### Author Response · Authors · 2024-08-13
> > >
> > > Thank you for your continued engagement with our paper and maintaining your positive review of the work. We are happy to hear that our answer to reviewer 9EgD with regards to the multidimensional $\lambda$ and our discussion regarding applications to LLMs was convincing. Thank you again.

---

### Official Review · Reviewer_9EgD · 2024-07-30

**Soundness:** 2
**Presentation:** 3
**Contribution:** 2
**Rating:** 4
**Confidence:** 4

**Summary:**

This paper proposes a method to post-process a pre-trained model to align with a subset of hypotheses on which the specific desired behaviors can be attained. The proposed method relies on proximity oblivious testers to give detection for the misalignment, based on which a conformal risk control process is used to calibrate the prediction interval to guarantee the farness to the desired subset of hypotheses.

**Strengths:**

1. The considered problem is a general and important. The properties to be aligned can include many possible instances.
2. The paper is overall well written and easy to follow.

**Weaknesses:**

1. The proposed aligning method is largely built on conformal risk control to calibrate the prediction interval, so the original technical contribution needs to be highlighted.
2. The generalization to multi-dimension conformal risk control is a bit straightforward and an immediate result from the original conformal risk control. The technical challenges need to be highlighted also.
3. The proposed method depends on the reliability of the POT, as mentioned in Line 181. However, to guarantee a rigorous prediction, it is important to consider the failure and errors from POT and how to guarantee the valid prediction in this case in details.

**Questions:**

N.A.

**Limitations:**

N.A.

---

> ### Author Rebuttal · Authors · 2024-08-06
>
> We would like to thank the reviewer for their time and effort in reviewing our paper and providing valuable feedback. We appreciate that the reviewer acknowledges the generality and importance of the problem we considered, as well as the flexibility of our approach concerning the properties that can be addressed. Below, we provide responses to each of the three listed weaknesses.
>
> The first listed weakness:
> “1. The proposed aligning method is largely built on conformal risk control to calibrate the prediction interval, so the original technical contribution needs to be highlighted.”
>
> We appreciate the call to better highlight our technical contribution. An important contribution of this work is combining two unrelated topics of conformal risk control and property testing to tackle the alignment problem. To the best of our knowledge, this is the first. We feel this methodological approach, along with our main theorem, is a strong contribution. We also want to bring to the review’s attention our technical contributions in Section 5. We bring attention to these results in our global response, but repeat here for convenience.
>
> Our theoretical results in this section build on random feature model theory (Mei and Montanari 2023, Misiakiewicz and Montanari 2023) to obtain insights regarding the impact of model size and sample on adherence to properties $\mathcal{P}$. We show that even if the true data generating process adheres to $\mathcal{P}$ if there is small noise bias then overparameterized models will fail to satisfy a desired property $\mathcal{P}$ regardless of how much data is collected. This result has significant implications with respect to the persistent need for alignment techniques regardless of how much training data large models are trained on.
>
> We hope this clarifies the strength of the technical contributions of our paper.
>
> The second listed weakness:
> 2. The generalization to multi-dimension conformal risk control is a bit straightforward and an immediate result from the original conformal risk control. The technical challenges need to be highlighted also.
>
> We agree with this point; we attempted to clearly state in both the proposition statement in the main text and the proof in the appendix that the proof closely mimics the original result.
>
> However, we do feel that the literature has overlooked the potential of something like multi-lambda conformal risk control so far, even in instances in which it would have been helpful. In particular, as mentioned in the global response, Yadkori et al. (2024) utilize two separate conformal procedures to calibrate two separate functions, which could have been simultaneously calibrated in tandem using multi-lambda conformal risk control. Other works, such as “Conformal Risk Control for Ordinal Classification” (Xu, 2023), assume that lower and upper bounds on a prediction set are given by functions l(lambda) and u(lambda) but requiring both ends of the interval to grow simultaneously may not be generally necessary; multi-lambda conformal risk control provides a flexible implementation of the conformal risk control algorithm in such settings.
>
> The third weakness:
> 3. The proposed method depends on the reliability of the POT, as mentioned in Line 181. However, to guarantee a rigorous prediction, it is important to consider the failure and errors from POT and how to guarantee the valid prediction in this case in details.
>
> We believe there is room for clarification on this point. While for general property testing algorithms, there may be two-sided errors, for POTs, we can guarantee that if $f$ satisfies property $\mathcal{P}$, then with probability one, the POT will correctly accept this function. In other words, there can be no false negatives or type II errors. The POTs can indeed have a false positive or type I error. This does not impact our main result, however, since for any valid testing algorithm (which we assume $T$ is), any function that $T$ accepts on at least $1-\alpha$ fraction of points is by definition as most $\alpha$ far from having the property that $T$ tests for.
>
> It is possible, however, for future work, that results concerning the distribution of the loss functions on test points (since here we only have a guarantee a bound on the expectation of the loss) may need to take into account the error probability of the tester. We thank the reviewer for bringing up this point, as it is closely aligned with ideas on our minds for potential future directions.
>
> We hope that our responses to each of these listed weaknesses sufficiently address the reviewer’s concerns and can convince them of the work’s contribution. As the reviewer mentioned, the address problem of this work is both general and interesting, and our approach is powerful and flexible. We hope the reviewer is more convinced that the approach is also theoretically and technically strong regarding both the new approach to alignment and the results regarding the persistent need for alignment based on the random feature model theory. Again, we would like to thank the reviewer for engaging with our work and hope their judgment of the work has become increasingly positive.

---

> > ### Author Response · Authors · 2024-08-13
> >
> > We would like to thank the reviewer again for their initial review and feedback on our paper. We hope that the reviewer may find our provided discussions and clarifications within our rebuttal convincing, but we would also be happy to address any further questions the reviewer may have regarding our paper or our rebuttal. Thank you again for your time and effort.

---

### Author Rebuttal · Authors · 2024-08-05

We thank all reviewers for their positive feedback and critical assessment of our paper. We aim to address remaining concerns and reinforce the contributions of our work. This response focuses on key points that may have been underemphasized, aiming to further convince the reviewers who gave us Accept (7) scores of our work’s quality and persuade those who gave us Borderline reject (4) scores to reconsider positively.

Reviewers appreciated the flexibility and applicability of our approach to various problems, noting its potential for domains such as generative AI, physical applications, and fairness. There were questions, however, around our focus on monotonicity and how our approach extends to generative AI. We want to emphasize that our methodology allows us to define and align desirable properties of generative models as long as we have a way to test for these properties at a per-input level.

We wish to highlight this claim via a section of the paper that may have been too buried originally. The connection to Yadkori et al. (2024) within Section 6 Related work was not intended only for paying correct dues to their work, but also to highlight the applicability of our methodology to a generative AI setting. We discuss that their use of conformal risk control to mitigate LLM hallucination can be captured as a specific case of our more general property alignment perspective.

We discuss there in Section 6 that “not hallucinating” can be considered the property of the function $(a, f)$, where $a$ determines whether the model should abstain, and if not, the function $f$ gives the output. This example also shows how our approach of forming the conformal intervals is not limited to subsets of the reals since, in this application, the conformal set is either just $f(X)$ or $\\{f(X), \text{Abstain}\\}$ for sufficiently conservative $\lambda$. We also highlight the use of multi-lambda conformal risk control at the end of Section 6, since Yadkori et al. (2024) use a separate conformal procedure for calibrating their match function and abstention function, but using multi-lambda conformal risk control and our methodology would allow these functions to be calibrated in tandem.

Including a generative AI experiment is beyond this paper’s scope, but our discussion highlights our approach's potential. Our methodology enables defining and aligning generative models' desirable properties if testable at a per-input level.

A significant technical contribution, perhaps underemphasized, is in Section 5. Our theoretical results in this section build on random feature model theory (Mei and Montanari 2023, Misiakiewicz and Montanari 2023) to obtain insights regarding the impact of model size and data quality and volume on adherence to properties $\mathcal{P}$. We show that even if the true data generating process adheres to $\mathcal{P}$, small noise bias will cause overparameterized models to fail to satisfy the desired property $\mathcal{P}$, regardless of how much data is collected for training. This result has significant implications: even if we keep making models bigger and use unlimited training data, we still need alignment techniques because training data always has some small imperfections.

Finally we want to share in this global response an additional experiment that we hope hammers home the general applicability of our method. We consider the “box” or sokoban environment [1] as presented in [2,3]. In this sokoban environment, of which we attached an image in the pdf, the Agent is rewarded for reaching the goal state in as few moves possible (-1 reward per move, 50 reward and termination of episode upon reaching goal state). There is also a box, and if the agent moves in the direction of the box the box is pushed in that direction. We train a tabular Q-learning agent for this goal, which expectedly converges to the optimal performance of 45.

However, the authors of [2] consider the case in which we wish to penalize the agent from committing an “irreversible” action, which in this case is an action that pushes the box into a corner. They add a penalty term to the reward that punishes deviations from the initial environment’s trajectory to account for this.

We use our alignment via conformal risk control to approach this safety problem. Our pretrained tabular Q-learning agent was not trained for side effects. Here, $f$ is $Q(s,a)$ mapping state, action pairs to q-values. The property of interest is whether it induces irreversible actions—whether the $\text{argmax}_a q$ value action for a state is irreversible.
Our calibration set is $X_i = s_i$ and $Y_i = 1$ if $a_i = \text{argmax}_a Q(s_i,a)$ is irreversible, 0 otherwise. The loss function for CRC is 0 if the highest q-value action is reversible or if the interval around the highest q value includes a reversible action’s q-value, otherwise 1.
Using CRC, we find optimal lambda pairs, with $\lambda^+=15.3$, $\lambda^-=0$. We provide a q-values distribution in the attached pdf. Our agent chooses the action with the max q-value if its interval is disjoint from others; otherwise, we defer to a safe action checker, choosing the highest q-value among safe actions. This approach is akin to the abstention problem in Yadkori et al. (2024), allowing the pretrained model to output its optimal action unless deemed too risky, in which case it checks for safe actions. Through our alignment via conformal risk control procedure we obtain an agent that achieves the optimal score of 43 while avoiding potential side-effects.

[1] AI safety gridworlds. J Leike et al.

[2] Penalizing side effects using stepwise relative reachability. V Krakovna, L Orseau, R Kumar, M Martic, S Legg

[3] Avoiding side effects by considering future tasks. V Krakovna, L Orseau, R Ngo, M Martic, S Legg - Advances in Neural Information Processing Systems, 2020

---

### Decision · Program_Chairs · 2024-09-25

**Decision:**

Accept (poster)

**Comment:**

This work discusses model alignment, and suggests a new frameworks towards doing so by combining ideas from property-testing and conformal risk prediction. And this idea seems novel and interesting as mentioned by some of the reviewers.
Nevertheless, the writing and clarity of the manuscript need to be improved as per the comments of the reviewers; and the technical novelty is quite mild. Nevertheless, conceptually the paper does bring novelty, and may very well spark new research directions.
From these reasons I view the paper as borderline, with tendency to accept.